# Establishment of regions of genomic activity during the *Drosophila* maternal to zygotic transition

Xiao-Yong Li[1], Melissa M Harrison[2], Jacqueline E Villalta[1], Tommy Kaplan[3]*[†], Michael B Eisen[1,4,5,6]*[†]

[1]Howard Hughes Medical Institute, University of California Berkeley, Berkeley, United States; [2]Department of Biomolecular Chemistry, University of Wisconsin, Madison, United States; [3]School of Computer Science and Engineering, The Hebrew University of Jerusalem, Jerusalem, Israel; [4]Department of Molecular and Cell Biology, University of California, Berkeley, Berkeley, United States; [5]Department of Integrative Biology, University of California, Berkeley, Berkeley, United States; [6]QB3 Institute, University of California, Berkeley, Berkeley, United States

**Abstract** We describe the genome-wide distributions and temporal dynamics of nucleosomes and post-translational histone modifications throughout the maternal-to-zygotic transition in embryos of Drosophila melanogaster. At mitotic cycle 8, when few zygotic genes are being transcribed, embryonic chromatin is in a relatively simple state: there are few nucleosome free regions, undetectable levels of the histone methylation marks characteristic of mature chromatin, and low levels of histone acetylation at a relatively small number of loci. Histone acetylation increases by cycle 12, but it is not until cycle 14 that nucleosome free regions and domains of histone methylation become widespread. Early histone acetylation is strongly associated with regions that we have previously shown to be bound in early embryos by the maternally deposited transcription factor Zelda, suggesting that Zelda triggers a cascade of events, including the accumulation of specific histone modifications, that plays a role in the subsequent activation of these sequences.

**\*For correspondence:** tommy@cs.huji.ac.il (TK); mbeisen@berkeley.edu (MBE)

[†]These authors are co-senior authors

**Competing interests:** The authors declare that no competing interests exist.

**Reviewing editor**: Robb Krumlauf, Stowers Institute for Medical Research, United States

## Introduction

In most animals, the first phase of embryonic development depends solely on maternally deposited proteins and RNAs and is often accompanied by very low or undetectable transcription (*Newport and Kirschner, 1982a*, *1982b*; *Tadros and Lipshitz, 2009*). After several hours to several days, depending on the species, zygotic transcription initiates, marking the beginning of a process known as the maternal-to-zygotic transition (MZT) during which maternally deposited RNAs are degraded and the zygotic genome assumes control of its own mRNA production.

In *Drosophila melanogaster*, sustained zygotic transcription begins around mitotic cycle 7, about an hour into development, although there is growing evidence that very low levels of transcription occur even earlier (*Ali-Murthy et al., 2013*; *ten Bosch et al., 2006*). Zygotic transcription gradually increases with each subsequent mitotic cycle, but it is not until the end of mitotic cycle 13 that widespread zygotic transcription is observed (*Pritchard and Schubiger, 1996*; *Lécuyer et al., 2007*; *Lott et al., 2011*; *McKnight and Miller, 1976*). This zygotic genome activation, along with the elongation of mitotic cycle, and cellularization of the syncytial nuclei defines the mid-blastula transition (MBT). Approximately 3000 genes are transcribed in the cellular blastoderm (*De Renzis et al., 2007*; *Lécuyer et al., 2007*; *Lott et al., 2011*). Of these, roughly 1000 are expressed in spatially restricted patterns

**eLife digest** For a fertilized egg to develop into an embryo, many genes must be switched on and off at specific times. A fertilized egg (or zygote) contains genetic material from both parents; and the life of the fruit fly *Drosophila melanogaster* begins with the nuclei that contain this genetic material repeatedly dividing for the first 2 hr. These nuclear divisions are initially controlled by molecules that the mother deposits into the egg cell. However, as these molecules degrade, the zygote's genome is activated and its own genes take control of embryonic development, in a process referred to as the 'maternal-to-zygotic transition'.

In the fruit fly zygote, this burst of regulated gene activation is likely to be accompanied by changes to the way that the DNA is packed inside the nuclei. Most DNA in a cell is packaged into a structure called chromatin, which can be marked at specific sites by chemical modifications. For example, chromatin can be acetylated or methylated, which alters its physical structure, helping the underlying genes to be either activated or repressed.

In the fruit fly, the first genes to be switched on (as well as many early developmental genes) have a DNA motif that is recognized, and is bound by, a protein called Zelda. The Zelda protein plays a major role in activating the genome of the early fruit fly embryo, by marking thousands of genes and regulatory regions for activation. This is somewhat similar to the activity of so-called 'pioneer' factors that alter chromatin structure to allow particular genes to be switched on or off, and to trigger the formation and development of specific tissues.

Here, Li et al. have investigated whether the Zelda protein—like known pioneer factors—also affects chromatin during the maternal-to-zygotic transition. Different chromatin modifications across the whole fruit fly genome were characterized at specific time-points during the maternal-to-zygotic transition, and the information gathered was then analyzed along with previous data on gene activity.

In the early stages of the maternal-to-zygotic transition, Li et al. found very few of the chromatin features that characterize more mature cells. This indicates that the chromatin is in a so-called 'naïve' state. As the transition progresses, Li et al. observed that the chromatin becomes acetylated before it is methylated, and that marks associated with activation appear before those associated with repression. Chromatin acetylation was strongly associated with the early binding of the Zelda protein to its target genes.

Li et al.'s findings show when, and in what order, the different features of mature chromatin appear in *Drosophila* zygotes. A future challenge will be to identify whether Zelda directly recruits the proteins that cause chromatin acetylation, or whether it blocks the changes to chromatin that repress gene expression.

---

(*Tomancak et al., 2007*; *Combs and Eisen, 2013*), a result of the differential binding by around 50 spatially patterned transcription factors to several thousands known and putative patterning transcriptional enhancers (*Li et al., 2008*; *MacArthur et al., 2009*).

In the cellular blastoderm, active genomic regions are biochemically distinct from the rest of the genome: they have relatively low nucleosome densities; are bound by transcription factors, polymerases and other proteins that mediate their activity; and have characteristic histone modifications (*Li et al., 2011*; *Nègre et al., 2011*). This high level of activity and relatively complex landscape of genome organization is remarkable given that an hour earlier the genome was being continuously replicated and doing little else. Although the *Drosophila* cellular blastoderm is among the most well-characterized animal tissues, the transition from quiescent to active state that precedes the formation of this tissue remains poorly understood, despite increasing evidence of its importance (*Liang et al., 2008*; *Blythe et al., 2010*; *Harrison et al., 2011*; *Nien et al., 2011*; *Liang et al., 2012*; *Lee et al., 2013*; *Leichsenring et al., 2013*).

We have previously shown that a single seven base-pair DNA motif is found in the vast majority of patterning enhancers active in the cellular blastoderm (*Li et al., 2008*), and that the maternally deposited transcription factor Zelda (ZLD) (*Liang et al., 2008*), which binds to this sequence, is present at these enhancers by mitotic cycle 8, in the early phase of the MZT (*Harrison et al., 2011*). ZLD binding sites were also found in the promoters of most genes activated in this early phase (*ten Bosch et al., 2006*), suggesting that ZLD may play a broad role in early embryonic genome activation and

suggesting that ZLD might play a role analogous to the pioneer transcription factors that choreograph the reorganization of genome activity during differentiation, as reviewed in *Zaret and Carroll, (2011)*.

Although ZLD mutants alter the expression of a large number of cellular blastoderm genes (*Liang et al., 2008*), and affect transcription factor binding in the cellular blastoderm (*Yanez-Cuna et al., 2012*; *Foo et al., 2014*; *Xu et al., 2014*), little is known about its molecular function or when its activity is required. We hypothesized that ZLD might affect transcription factor binding and enhancer activity indirectly through interactions with chromatin. To further explore this possibility, and to better situate ZLD action in the broader context of early embryogenesis, we decided to characterize the chromatin landscape of *D. melanogaster* embryos throughout the MZT.

## Results

### Quantitative mapping of nucleosome occupancy and histone modifications across the MZT

To define the chromatin landscape before, during and after the maternal-to-zygotic transition, we collected *D. melanogaster* (Oregon-R) embryos from population cages at 25°C for 30 min, and aged them for 55, 85, 120 and 160 min to target mitotic cycles 8, 12, 14a and 14c respectively, prior to fixing them with formaldehyde (*Figure 1A*).

As *D. melanogaster* females often retain eggs post-fertilization, leading to unacceptable levels of contaminating older embryos in embryo pools (*Harrison et al., 2011*), we manually removed embryos of incorrect stages by inspection under a light microscope, as previously described (*Harrison et al., 2011*). The purity of the resulting embryo pools was confirmed by examining the density of nuclei in 4',6-diamidino-2-phenylindole (DAPI) stained samples from each pool (*Figure 1B,C*).

We carried out chromatin immunoprecipitation and DNA sequencing (ChIP-seq) using commercial antibodies against nine post-translation modifications (acetylation at H3K9, H3K18, H3K27, H4K5 and H4K8, mono-methylation at H3K4, and tri-methylation at H3K4, H3K27 and K3K36), as well as histone H3 (*Table 1*).

As we sought to compare not just the genomic distribution of marks but also their relative levels across the MZT, we developed a strategy to normalize across time-points for the same antibody. Briefly, we prepared chromatin from stage 5 *Drosophila pseudoobscura* embryos (mitotic cycle 14), and 'spiked in' a fixed amount of this common reference to each *D. melanogaster* chromatin sample prior to ChIP and sequencing. We chose *D. pseudoobscura* since it is sufficiently diverged from *D. melanogaster* that there is very little ambiguity in the assignment of reads to the correct species (*Paris et al., 2013*).

The *D. pseudoobscura* chromatin served as an internal standard. Since the *D. pseudoobscura* chromatin in each sample was identical, we expected it to be identically immunoprecipitated (within experimental error). Indeed, we found that both the number of peaks (*Figure 1—Figure supplement 1A*) and the peak-by-peak signal (*Figure 1—Figure supplement 1B*) for the *D. pseudoobscura* fraction across time-points were fairly stable. We therefore used the relative recovery of *D. melanogaster* compared to *D. pseudoobscura* in each time-point as a measure of the relative abundance of the corresponding mark in that time-point.

### Dramatic shift in chromatin during the maternal-to-zygotic transition

We used three measures of genome-wide recovery of each histone mark to examine their dynamics: the total normalized number of *D. melanogaster* reads (*Figure 2A*), the number of regions scored by MACS as enriched (*Figure 2B*) and the average ChIP signal among all enriched regions (*Figure 2C*). These all gave qualitatively similar results except for H4K5ac, which had anomalously few peaks at early stages despite being found at uniformly high levels across the genome.

As expected, global levels of histone H3 were relatively stable, although we observed a gradual increase of approximately 1.4-fold over time, possibly reflecting an overall increase in nucleosome density and chromatin compaction in cycle 14 relative to cycle 8. The replication associated mark H4K5ac (*Sobel et al., 1994*), found ubiquitously across the genome, declined rapidly from cycle 8 onwards, consistent with the elongation of cell cycles duration over time and the decreasing fraction of nuclei caught in S phase. The remaining marks all showed dramatic increases over the MZT. H4K8ac, H3K18ac, and H3K27ac were enriched at hundreds of loci at cycle 8 and steadily increased through cycle 14. The remaining marks, H3K9ac, H3K4me1, H3K4me3, H3K36me3 and H3K27me3, were

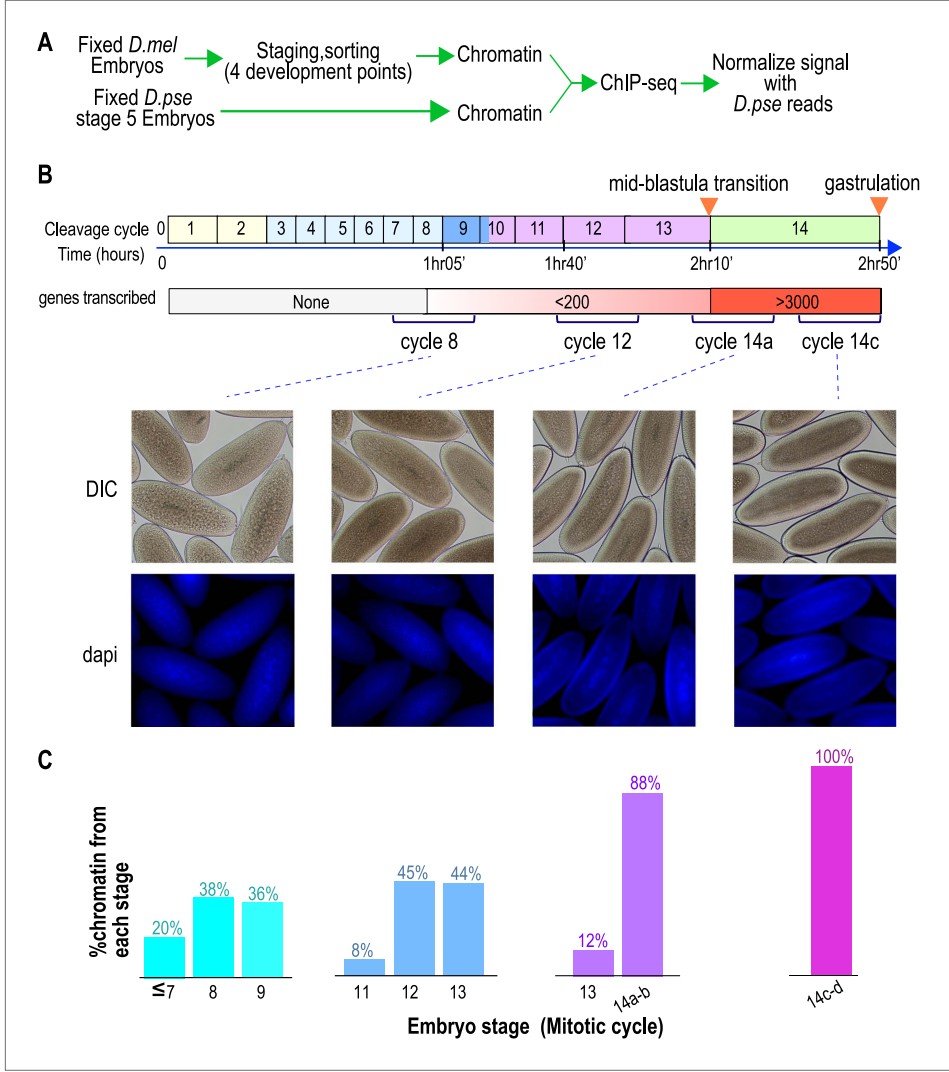

**Figure 1**. Hand sorting based on morphology results in tightly staged embryos. (**A**) Experimental scheme. *D. melanogaster* embryos were collected and allowed to develop before being fixed with formaldehyde. Fixed embryos were hand sorted to obtain pools of embryos within a relatively narrow age distributions between mitotic cycle 8 and the end of cycle 14. To serve as carrier and normalization standard, chromatin from fixed stage 5 (cycle 14) *D. pseudoobscura* embryos was prepared and added to the chromatin from the sorted embryos prior to chromatin immunoprecipitation. In ChIP-seq data analysis, the sequencing reads for *D. pseudoobscura* were used to normalize the *D. melanogaster* ChIP-seq signals. (**B**) Embryo collection and sorting. The timeline of the early embryogenesis is depicted on top with the relative lengths of each mitotic cycle approximated by the size of the box. The developmental stages (from 1–5) are indicated by different colors. The earliest sustained transcription is detected is at cycle 7, and the mid-blastula transition (MBT) occurs when a large number of genes are transcriptionally activated at approximately the end of cycle 13. We generated four pools of sorted embryos with developmental stages centered around cycles 8, 12, 14a, or 14c as shown by differential interference contrast (DIC) and DAPI. (**C**) We determined the distribution of the developmental cycle of the embryos in each pool as shown by counting the number of nuclei in DAPI-stained embryos or by examining the extent of membrane envagination during cycle 14.

The following figure supplement is available for figure 1:

**Figure supplement 1**. Normalization using *D. pseudoobscura*.

effectively absent at cycles 8 and 12, but showed sharp increases at cycle 14a. This distinction between these two groups of marks is evident when examining levels of histone modification at individual loci (***Figure 3***).

**Table 1.** Antibodies used in this study

| Mark | AB source | AB catalog # |
|---|---|---|
| H3 | Abcam | ab1791 |
| H4K5ac | Millipore | 07-327 |
| H4K8ac | Abcam | ab15823 |
| H3K18ac | Abcam | ab1191 |
| H3K27ac | Abcam | ab4729 |
| H3K4me1 | Abcam | ab8895 |
| H3K4me3 | Abcam | ab8580 |
| H3K9ac | ActiveMotif | 39,138 |
| H3K27me3 | Millipore | 07-449 |
| H3K36me3 | Abcam | ab9050 |

## Chromatin changes in transcribed regions are associated with gene activation

The transcription of several thousand genes is initiated during the period covered by our analyses, and we were interested in the relationship between the timing of the onset of transcription at individual loci and their chromatin dynamics. We used high-temporal resolution expression data previously collected by our lab (*Lott et al., 2011*) to identify genes that were exclusively zygotically transcribed, genes whose mRNAs were deposited into the egg maternally, and genes that are not transcribed in the early embryo. We divided the exclusively zygotic genes into four temporal groups according to their onset times (*Figure 4*), and used RNA polymerase II binding data from (*Chen et al., 2013*) to divide maternally deposited genes into those transcribed in the early embryo (maternal-zygotic genes) and those that are not. We then examined patterns of nucleosome enrichment and histone modifications around the transcription start sites, and in the gene body, of genes in each of these classes (*Figure 5*).

Nucleosome free regions (NFRs; areas of relatively low histone H3 recovery) emerged around the transcription start sites of zygotically transcribed genes at roughly the same time that their transcripts were evident in our transcription data (*Figure 5*). Several histone modifications appeared along with transcription: H4K8ac, H3K18ac and H3K27ac (*Figure 5*). In contrast, H3K9ac and the four histone methylation marks examined here were absent until cycle 14a, when widespread transcription begins.

In cycle 14 embryos, regions both upstream and downstream of the promoters of zygotically expressed genes were enriched for the Polycomb-associated mark H3K27me3 (*Figure 5*), while the mark was almost completely absent from maternal genes, consistent with the known role of Polycomb group proteins in cell-type specific silencing of developmental genes (*Boyer et al., 2006*; *Lee et al., 2006*; *Schwartz et al., 2006*).

We also observed that maternally deposited genes, both those transcribed in the early embryo and those that are not, had fairly strong NFRs upstream of the promoter at all time points (*Figure 5*). A strong transcription-independent NFR in maternally deposited genes was been previously described by *Gaertner et al., (2012)*, who also showed that, based on DNA sequence alone, these genes also have a strong predicted NFR upstream of the promoter. Our data extend this observation, showing that this NFR is developmentally stable. It has been previously observed that maternal-zygotic genes have different promoter motifs than zygotic genes (*Gaertner et al., 2012*; *Chen et al., 2013*) and this may be at least partially responsible for the difference.

## Dynamic histone marks at blastoderm enhancers

Many of the genes transcribed by cycle 14 are expressed in clear spatial patterns (*Lécuyer et al., 2007*; *Tomancak et al., 2007*; *Combs and Eisen, 2013*) driven by the action of distinct transcriptional enhancers. Although several catalogs of blastoderm enhancers exist (*Gallo et al., 2011*), they are limited in scope. To generate a larger set of likely enhancers, we took advantage of the strong correlation between the binding of transcription factors known to regulate blastoderm expression and enhancer activity (*MacArthur et al., 2009*; *Fisher et al., 2012*). We calculated the cumulative in vivo binding landscape of 16 early developmental transcription factors, including the anteroposterior regulators Bicoid, Caudal, Hunchback, Giant, Krüppel, Knirps, Huckebein, Tailless, and Dichaete; and the dorsoventral regulators Dorsal, Snail, Twist, Daughterless, Mothers against dpp, Medea, and Schnurri (*MacArthur et al., 2009*). We then identified a set of 784 regions showing the strongest overall binding, excluded peaks overlapping promoters and coding regions, and obtained a stringent set of 588 likely blastoderm enhancers in introns and intergenic regions.

The chromatin state we observed associated with these putative enhancers during cycle 14, when blastoderm enhancers are active and bound by multiple transcription factors were as expected based on previous studies of transcriptionally active mammalian and insect cells and

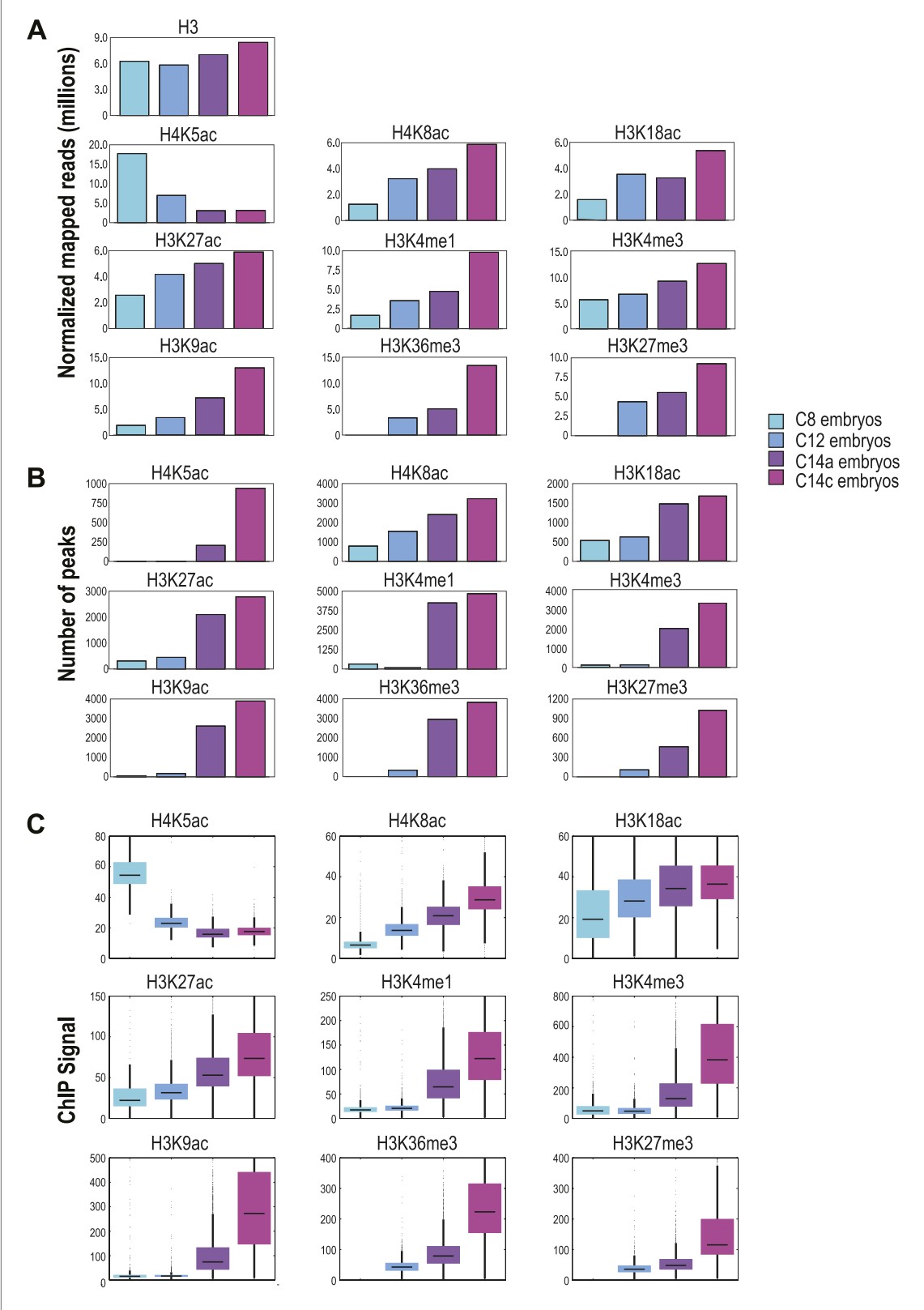

**Figure 2**. Global levels of histone marks change over early development. (**A**) The number of aligned reads (after normalization to *D. pseudoobscura*) for the four developmental time points are indicated for each histone mark and histone H3. (**B**) The number of peaks detected using the peak calling

*Figure 2. Continued on next page*

*Figure 2. Continued*

program, MACS (*Zhang Y et al., 2008*), for each histone mark at each stage are shown. (**C**) Box plots show the trend of average ChIP-seq signals over ±500 bp around the peaks detected across all stages for each histone mark. The dark line in the middle of the plot represents the median, the edges of the box represent the first and third quartiles.

tissues (*Heintzman et al., 2007*, *2009*; *Wang et al., 2008*; *Creyghton et al., 2010*; *Kharchenko et al., 2011*; *Nègre et al., 2011*; *Rada-Iglesias et al., 2011*). As shown in *Figure 6*, enhancers at cycle 14 exhibited strong nucleosome depletion (*Kaplan et al., 2011*; *Li et al., 2011*). Flanking nucleosomes were enriched with H3K4me1, H3K27ac, and H3K18ac, marks previously shown to be enriched at active enhancers (*Heintzman et al., 2007*; *Wang et al., 2008*; *Creyghton et al., 2010*; *Kharchenko et al., 2011*; *Rada-Iglesias et al., 2011*) and depleted for H3K4me3 and H3K36me3. As many early developmental genes are located in broad domains of Polycomb-associated H3K27me3

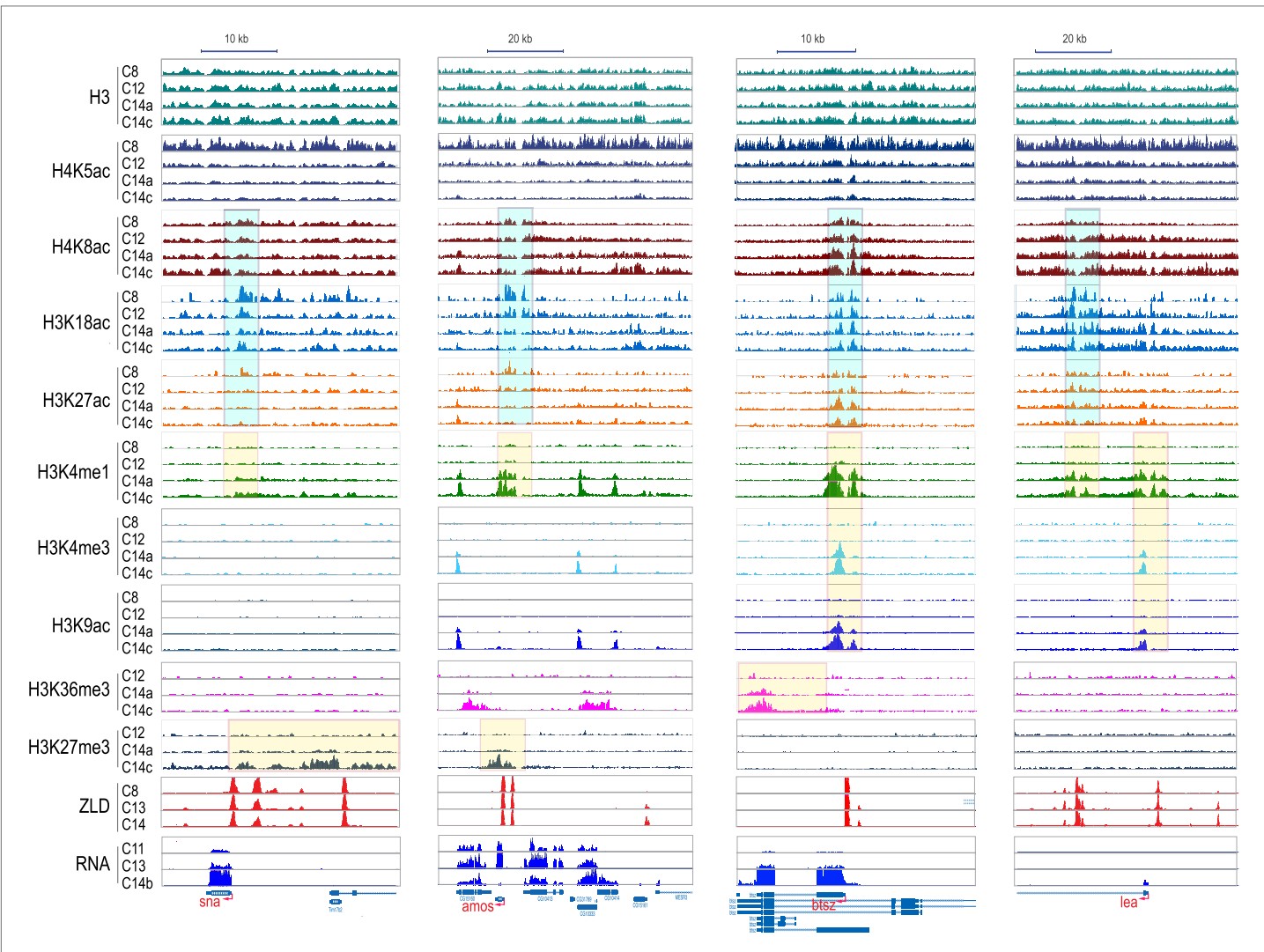

**Figure 3**. Dynamics of H3 and histone marks around selected genes. The normalized ChIP-seq signal profiles, for histone H3 and nine different histone marks at four development time points at selected genomic loci. Shown are the early onset genes, sna and amos, and late onset genes, btsz and lea. The peak regions of histone acetylation marks detectable prior to MBT are highlighted with cyan-colored boxes. The peak regions for histone marks detected only after the MZT are highlighted by yellow-colored boxes. Below are the ZLD ChIP-seq profile (*Harrison et al., 2011*) from c8, 13, and 14 embryos, as well as RNA-seq signals (*Lott et al., 2011*) at c11, c13, c14b.

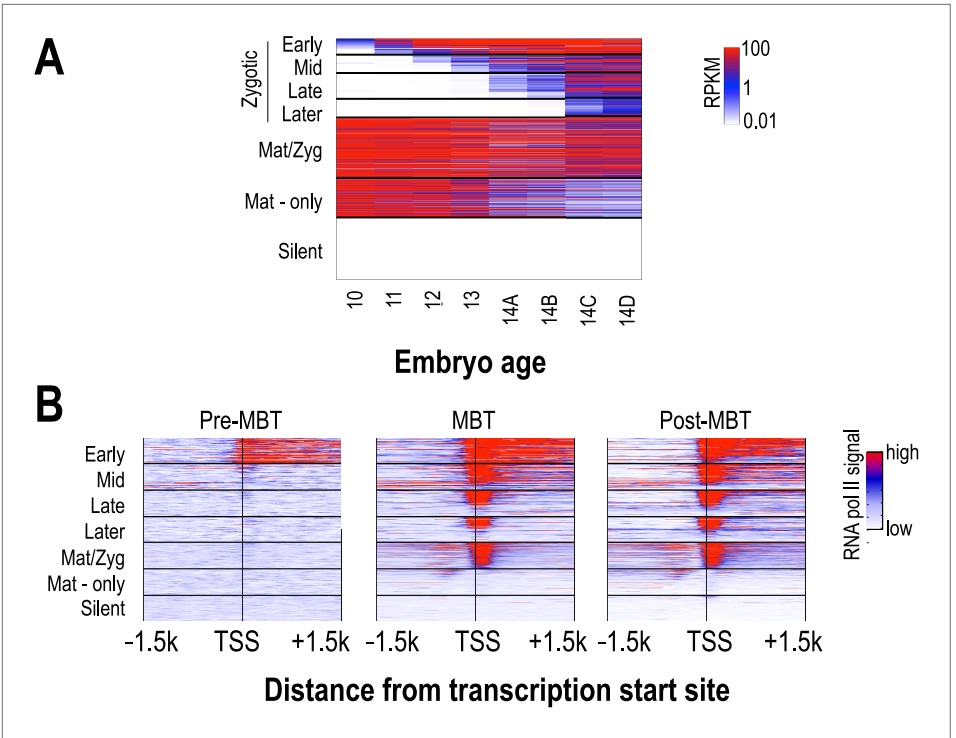

**Figure 4**. Classification of genes based on timing of transcriptional initiation during early embryogenesis. Using single-embryo RNA-seq data from our group (*Lott et al., 2011*), we identified three broad classes of genes: those at high levels in the earliest embryos ('maternal' genes), those not present in the earliest embryos, but transcribed prior to or during mitotic cycle 14 ('zygotic'), and those not present through cycle 14 ('silent'). We further divided the zygotic genes into four different groups based on their onset of zygotic expression—'Early' genes with onset of expression around mitotic cycles 10–11, 'Mid' genes at cycles 12–13, 'Late' genes at early cycle 14, and 'Later' zygotic genes whose onset of expression was during late cycle 14. Post-MBT polII ChIP data (*Chen et al., 2013*) was used to define two maternal groups of genes—those bound by polII in the embryo ('Mat/Zyg' genes), and those that are strictly maternally deposited ('Mat-only' genes). (**A**) Heatmap showing the expression levels for all groups at 8 timepoints (from cycle 10 through 14D) across the MZT (*Lott et al., 2011*). (**B**) Heatmaps showing RNA polymerase II ChIP-seq signals (*Chen et al., 2013*) around the transcription start sites (±1.5 kb) of the genes in each category for three developmental time points, pre-MBT (left), MBT (middle), and post-MBT (right) embryos. Genes within each group were ordered based on cycle 14 RNA polymerase II signals (genes with the highest signal are on top).

(*Nègre et al., 2011*), many of our putative enhancers are found within large regions containing high levels of this repressive mark.

While the chromatin status of enhancers at cycle 14 has been intensively investigated, their status earlier during the MZT has received much less attention. Our set of likely blastoderm enhancers had, as a class, relatively high nucleosome densities at mitotic cycle 8 (*Figure 6*; top left). At this early time point, flanking nucleosomes were weakly enriched for the three early appearing histone acetyl marks, especially H3K18ac, with these marks becoming more strongly enriched by cycle 12.

The process of nucleosome depletion was initially evident at cycle 12, but was much stronger at cycle 14a, when these enhancers begin to be active. The enhancer-associated mark H3K4me1 appeared on flanking nucleosomes by cycle 14a, but the repressive H3K27me3 did not appear in surrounding regions until cycle 14c. This raises the possibility that early events, reflected by the appearance of these enhancer-associated acetylation and methylation marks, play an important role in keeping these regions active once broader domains of inactivity are established.

## Early appearing chromatin features are associated with binding sites for the transcription factor ZLD

As expected from our previous study (*Harrison et al., 2011*), almost all enhancer sequences described above are also strongly associated with early binding of the transcription factor ZLD (*Figure 6*). This

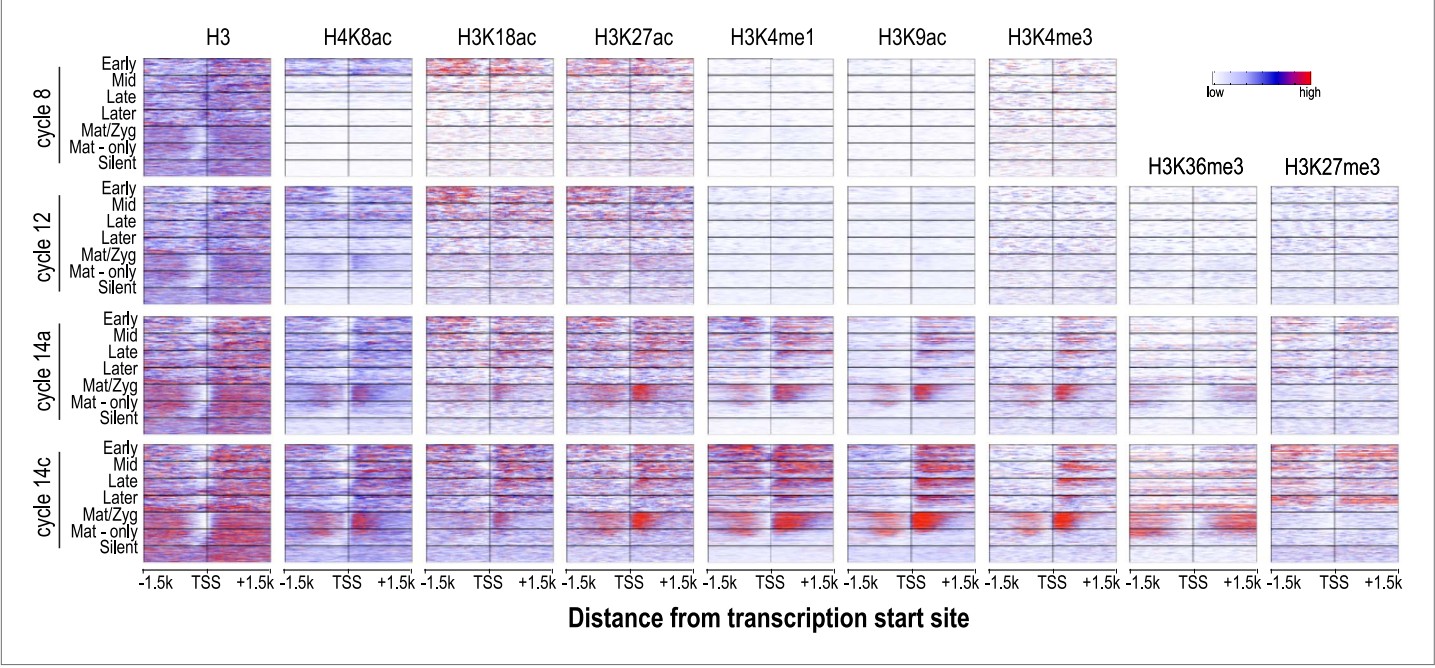

**Figure 5**. Relationship between H3 depletion, histone modifications and transcription dynamics. Heatmaps show ChIP-seq signals for histone H3 and different histone modification marks at each stage centered around the transcription start sites (±1.5 kb). Genes are groups and ordered as described in *Figure 4*. For each histone mark and for histone H3 the same color scaling was used for heatmaps across all four developmental time points. TSS = transcriptional start site.

suggests that at least at the enhancers, ZLD binding is likely to play a major role in directing the deposition of the histone acetylation marks in early embryos. To investigate this further and to identify other factors that may play a major role in determining overall histone acetylation patterns, not just the enhancers in early embryos, we carried out k-mer enrichment analysis and used the motif search tool MEME (*Bailey and Elkan, 1994*) to identify sequence motifs associated with different histone mark peaks identified at each stage. We found that the motif most strongly correlated with the early appearing marks, H3K27ac, H3K18ac and H4K8ac, was ZLD's CAGGTAG binding (*Figure 7A*). A small number of other motifs also showed modest enrichment using these two methods, but they failed to show substantial enrichment when the enrichment is plotted around the histone mark peaks. These analyses thus suggested a close connection between ZLD binding and early histone acetylation in general, which is further highlighted by the extremely high degree of overlap between early (cycle 8) ZLD-bound peaks and early (cycles 8 or 12) peaks for H3K27ac, H3K18ac and H4K8ac (*Figure 7B*). The relationship is quantitative, with higher levels of ZLD binding coupled to increased levels of the same three marks in cycle 8 and cycle 12 embryos (*Figure 7C*). The relationship between ZLD binding and these histone marks decays over time (*Figure 7D*), likely reflecting the increasingly complex transcriptional profile of the genome. However, the strength of this association in early stages of the MZT suggests that ZLD is indeed a dominant factor shaping the early chromatin landscape.

## Histone marks at enhancers are decreased in embryos lacking maternal ZLD

To directly analyze the role that ZLD plays in the activation of the zygotic genome during the MZT, we carried out a limited series of ChIP experiments using embryos lacking maternal *zld* mRNA, obtained from *zld-* germ-line clones (*Liang et al., 2008*). These females lay significantly less than their wild-type counterparts, and thus obtaining sufficient amounts of staged chromatin was a challenge.

ChIP with an anti-ZLD antibody on these nominally *zld-* embryos at mitotic cycle 12 to mid-14, showed modest ZLD binding at the same set of sites bound in wild-type embryos (*Figure 8*). This residual ZLD activity is likely due to weak zygotic transcription of the paternal copy in female embryos

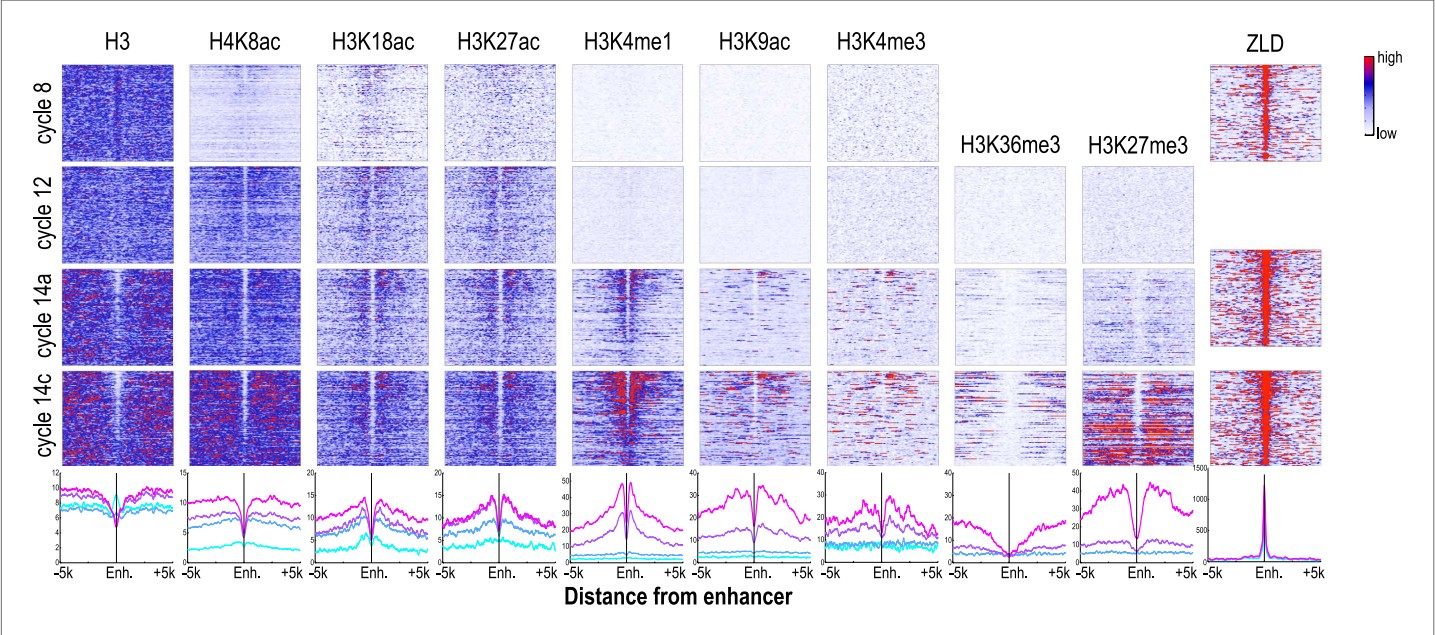

**Figure 6**. Dynamics of histone H3 depletion and histone modifications around blastoderm embryo enhancers. Heatmaps show ChIP-seq signals for histone H3 and different histone modification marks at each stage centered around putative enhancers (as described in text). Enhancers are ordered by chromatin accessibility, as measured by DNaseI–seq signals from cycle 14 embryos (**Thomas et al., 2011**) from high (top) to low (bottom). On the right, the heatmaps show the ChIP-seq signals for ZLD binding around these enhancers at c8, c13, and c14 (**Harrison et al., 2011**). Line plots at the bottom show the average ChIP-seq for histone H3, histone modifications, and ZLD at each stage around the enhancers. Enh. = enhancer.

(*zelda* is on the X chromosome and thus male offsprings do not receive a functional *zelda* from their father). Thus these ChIP data reflect the depletion, rather than complete elimination, of ZLD.

Intergenic ZLD-bound regions (**Figure 8B**) had a marked loss of H3K4me1 and a decrease of H3K18ac. The ZLD-associated NFR present in wild-type embryos was almost completely gone. In contrast, we saw a limited effect of ZLD depletion on the promoter histone state globally. There is still a strong NFR, and the marks we observed are present at roughly the same levels.

## Discussion

### Nature of chromatin changes during MZT

During the first stage of embryonic development, the genome must be reprogrammed from the differentiated states associated with the egg or sperm to create a set of totipotent cells capable of generating a new organism. By combining high-resolution gene expression analysis with precise mapping of nine histone marks throughout this early stage of development, our data suggest that this reprogramming, at least in *Drosophila melanogaster*, occurs by transitioning through a naïve state in which many histone marks commonly present in somatic cells are absent or at comparably low levels. We further demonstrate that histone acetylation of H3K18, H3K27, and H4K8 precedes most histone methylation. Thus we suggest that the establishment of the totipotent chromatin architecture proceeds in an ordered process with acetyl marks being deposited prior to methyl marks.

Studies in other organisms have similarly suggested that this early reprogramming is characterized by a transition through a relatively unmodified chromatin state. There is a loss and then reestablishment of DNA methylation following fertilization in mouse embryos (**Santos et al., 2005**). Immunostaining in mouse and bovine embryos has demonstrated that some histone methylation marks are removed following fertilization (**Burton and Torres-Padilla, 2010**). Additionally, studies in zebrafish have demonstrated widespread changes in chromatin marks as the embryo progresses through the MZT. While the extent and location of specific histone modifications in zebrafish is not consistent between recent studies (**Vastenhouw et al., 2010**; **Lindeman et al., 2011**), a general widespread increase in histone methylation (H3K4me3, H3K27me3, H3K36me3, and H3K9me3) is evident at the MZT. Thus in most, if

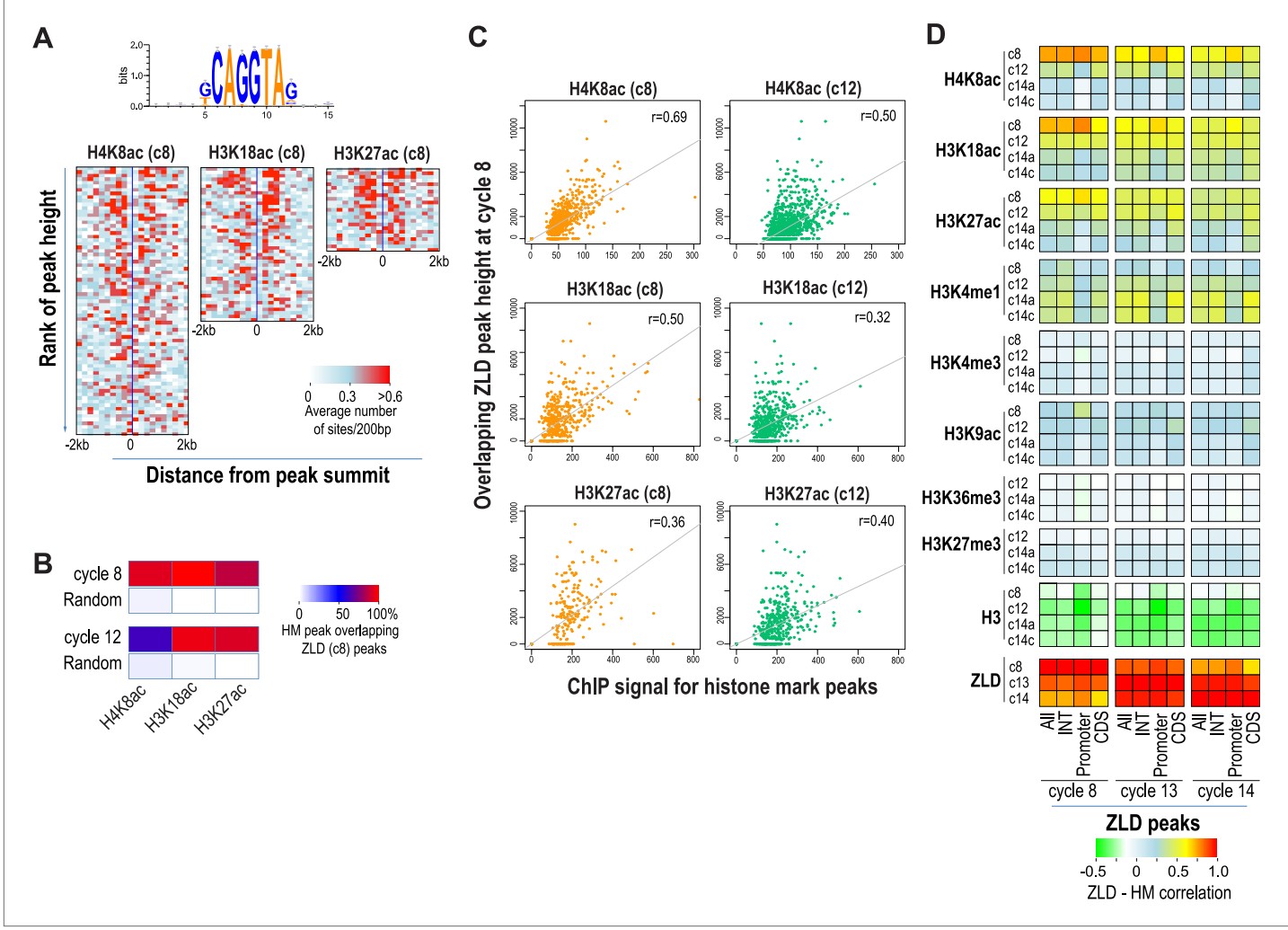

**Figure 7**. Relationship between histone occupancy, histone modification pattern and ZLD binding. (**A**) ZLD DNA binding motif enrichment around cycle 8 peaks for H4K8ac, H3K18ac, and H3K27ac. Peaks were ranked based on peak height, divided into bins of 100, and analyzed. Heatmaps show, for each location (column) and set of peaks (row), the average number of ZLD putative sites at each position/set. (**B**) Heatmap showing the overlap between histone acetylation peaks detected at cycle 8 and 12, a.nd ZLD peaks detected at cycle 8 (top 2000 ranked peaks). As a control, overlaps between histone mark peaks with random set of genomic positions that matched the number of ZLD peaks are shown. (**C**) Scatter plots showing the correlation between the signals around the peaks for the histone acetylation marks at cycle 8 and cycle 12 (X-axis) and the heights of the associated ZLD peaks within 1 kb of the histone mark peaks (Y-axis). The signal for each peak was the average over the ±1 kb region surrounding the peak. The correlation coefficient (r) for each plot is shown. (**D**) Heatmaps showing Pearson correlation coefficients between ChIP signal of top 5000 ZLD peaks and histone marks at same locations. ChIP-seq signals for histone H3 was averaged over a ±200 bp region around each ZLD peak. Histone marks were averaged over ±1 kb around ZLD peaks. The correlation coefficients were calculated individually for all the ZLD peaks ('All'), for intergenic and intronic ZLD peaks ('INT'), for promoter peaks ('Promoter'), and for ZLD peaks within coding sequences (CDS).

not all, organisms studied to date there is a dramatic increase in the abundance of histone modifications at the MZT, coinciding with zygotic genome activation.

One important lingering question is how this naïve state is established; whether there is a specific system that removes gametic marks associated with sperm and egg at fertilization, or whether the rapid replication cycles of early development are simply incompatible with active and differentiated chromatin. The lack of some histone marks in developing mouse and bovine embryos, which do not undergo rapid cell cycles early in development, suggests that while the mechanisms may vary between species, the removal of parental histone modifications may be a general feature of reprogramming. In the future, it will be important to understand how this transition is regulated to allow for the generation of a totipotent cell population.

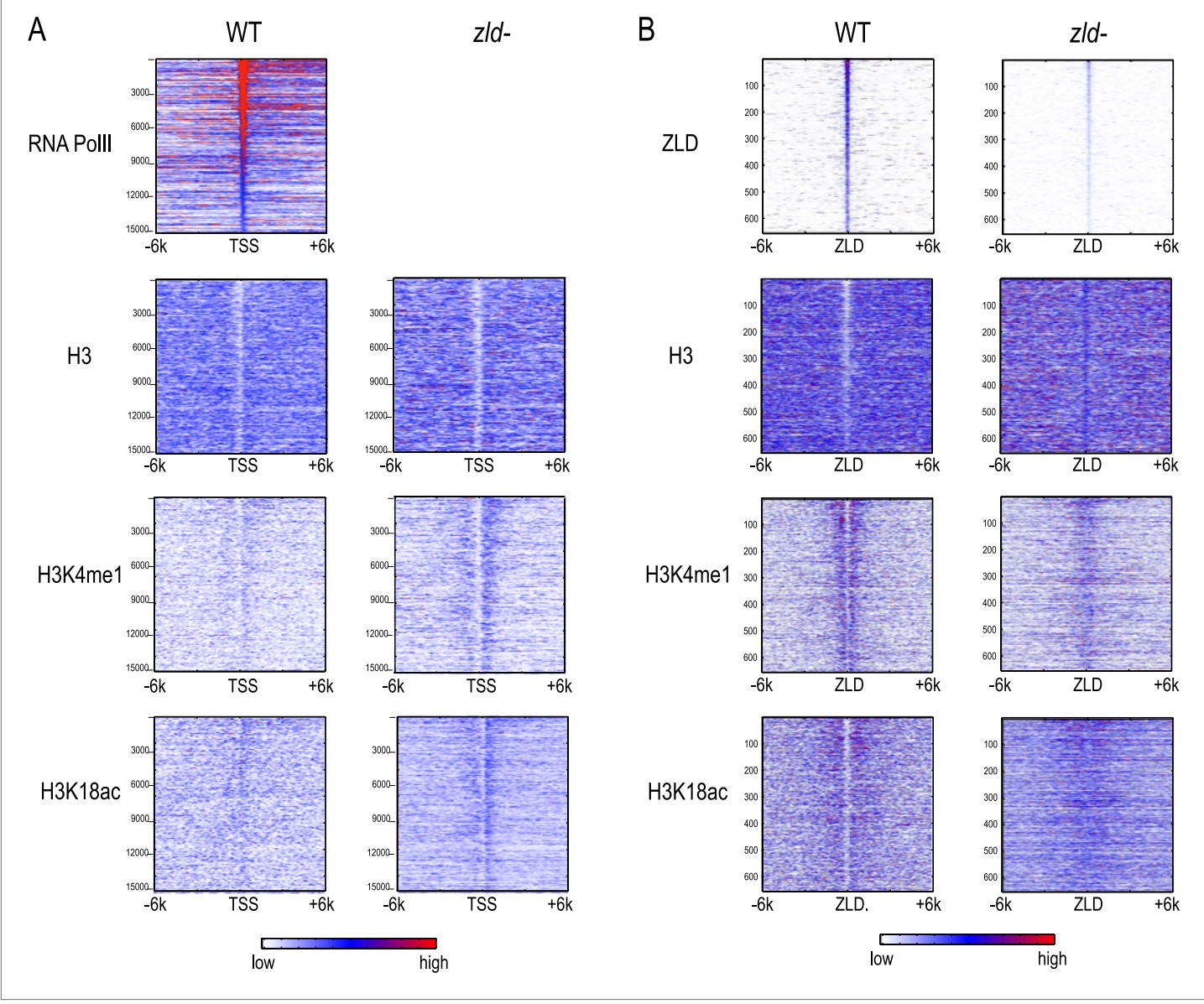

**Figure 8**. Effect of zld mutation on histone occupancy and modifications. Heatmaps show ChIP-seq data from WT embryos (left) and embryos lacking maternal *zld* (right). (**A**) Heatmaps centered at transcription start sites (TSS) and ordered by cycle 14 RNA polymerase II binding. (**B**) Heatmaps centered around intergenic ZLD peaks (*Harrison et al., 2011*). Shown are all the intergenic and intronic peaks, among top 1000 ZLD bound regions (total of 656), ordered by ZLD ChIP signal and aligned by peak position. TSS, transcription start site.

## Early transcription does not require marks canonically associated with activation

The presence of several post-translational modifications (e.g. H3K4me3 and H3K36me3) is correlated with transcription and with different states of gene expression in a variety of eukaryotic cells and tissues (c.f. (*Mikkelsen et al., 2007*)). By comparing high temporal-resolution transcription data to the quantitative histone modification data, we find that, at least in the early embryo, transcription often occurs in the near absence of these marks. These data confirm and extend the previous observation that H3K4me3 levels are first detectable at the MBT (*Chen et al., 2013*). Interestingly global H3K36me3 levels are also below the level of detection in the mouse embryo as it undergoes the first wave of zygotic genome activation (*Boskovic et al., 2012*).

Together these data suggest that marks canonically associated with gene activation at later stages of development are not associated with transcriptional activity in the very early zygote. Moreover, recent data has demonstrated that these marks may not be required for transcription later in development. For example, in the *Drosophila* wing disc, methylation of H3K4me3 is not required for transcriptional activity (**Hodl and Basler, 2012**). In *Drosophila* embryonic tissue culture cells transcriptionally active euchromatin can be divided into two classes and only one of these is enriched for H3K36me3 (**Filion et al., 2010**). Therefore transcription in the absence of H3K4me3 and H3K36me3 is likely not a distinctive feature of early embryonic development. Additionally, data from *Saccharomyces cerevisiae* show robust transcriptional activation of a gene localized to heterochromatin in the presence of minimal amounts of H3K36me3 (**Zhang et al., 2014**).

By contrast, in zebrafish H3K4me3 was identified at promoters of genes prior to their occupancy by RNA polymerase (**Vastenhouw et al., 2010**; **Lindeman et al., 2011**). Because those genes marked by H3K4me3 early are more likely than most genes to be activated at the MZT, it has been proposed that this mark is preparing genes for activation in the early zebrafish embryo. Thus, while histone marks can be associated with specific transcriptional outputs it appears that they are neither necessary nor sufficient for predicting gene expression.

## A simple model for early genome activation

While the data presented here are far from complete, together with our previously published high-resolution transcriptional analysis, they suggest a model in which regions of genomic activity in the cellular blastoderm are established by events that transpire earlier in development. In particular, they indicate that the binding of ZLD to target sites across the genome prior to the MZT may trigger a cascade of events—reflected in early histone depletion, the appearance of several histone acetylation marks, and the subsequent appearance of functional class-specific methylation marks—that may act to counter the establishment of Polycomb-mediated repression in many loci (**Figure 9**).

Indeed it has been shown in *D. melanogaster* S2 cells that acetylation of H3K27 by Nejire inhibits Polycomb silencing and the establishment of H3K27 trimethylation (**Tie et al., 2009**), and ZLD may play a role in the process by directing H3K27 acetylation to enhancers. One possibility is that ZLD directly recruits histone acetyltransferases, several of which including Nejire and Diskette are maternally deposited, and that these modifications play a direct or indirect role in genome activation. Alternatively, ZLD may simply act as a kind of steric impediment to subsequent chromatin compaction and silencing—with the observed histone acetylation an indirect byproduct of early ZLD binding.

We have previously observed that, while ZLD binding is fairly stable across the MZT, some of the regions it binds at cycle 8 are unbound at cycle 14 (**Harrison et al., 2011**). This may reflect the need for other factors to work in conjunction with ZLD while more restrictive chromatin is established. Indeed, while ZLD protein levels remain high through the MZT, the increasing number of nuclei means that absolute ZLD levels are dropping in each nucleus and may reach a point at which ZLD binding alone is insufficient to keep regions active or resist silencing.

## ZLD as a pioneer transcription factor

Work from Ken Zaret and others over the past decades has identified a class of transcription factors, known as 'pioneer' factors, that bind early to enhancers during differentiation and thereby promote the binding of other factors to the enhancer. Zaret attributes two characteristics to pioneer factors: 1) they bind to DNA prior to activation and prior to the binding of other factors, and 2) they bind their target sites in nucleosomes and in condensed chromatin (**Zaret and Carroll, 2011**).

ZLD clearly has the first characteristic. But it is not clear that is has the second. Our data suggest that there is essentially no condensed chromatin in the early embryo, as nucleosome density is relatively low, nucleosomes are relatively evenly distributed across the genome, and hallmarks of repressed chromatin are absent. This is consistent with the unusually broad binding of ZLD to its target sequences: ZLD binds to more than fifty percent of its target sites in the genome, far more than what is typical for other factors later in development (**Harrison et al., 2011**). We and others have shown that the restricted binding of other factors is largely due to the occlusion of most of their sites by condensed chromatin. Perhaps ZLD binds to a large fraction of its sites because there simply is no condensed chromatin in the early embryo. If so, ZLD would not require, and therefore would likely not possess, the ability to bind its sites in condensed chromatin.

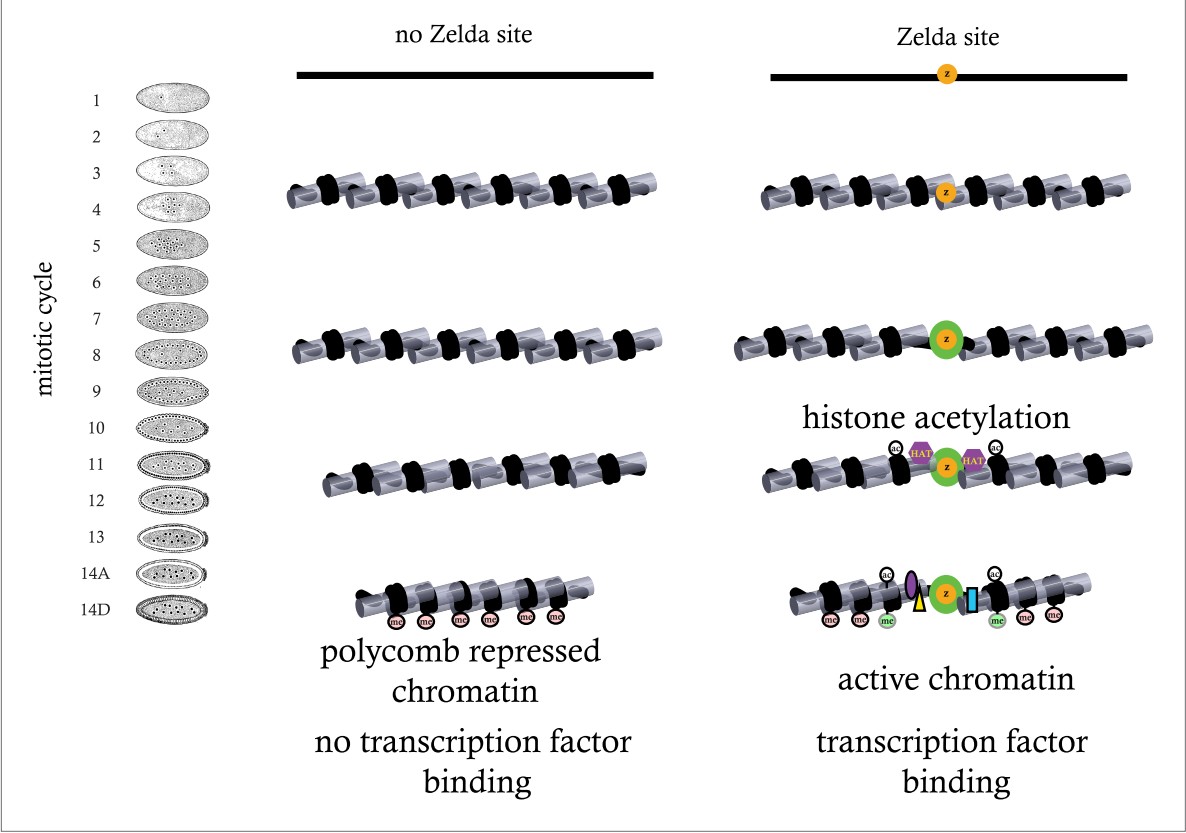

**Figure 9**. Model for ZLD function during zygotic genome activation. ZLD binds to enhancers in pre-MBT embryos at as early as cycle 8. This leads to histone acetylation and nucleosome remodeling around ZLD binding sites, which facilitates binding by other transcription factors, and in many other cases leads to additional deposition of histone marks including H3K4me1 while at the same time prevents local deposition of repressive histone mark H3K27me3 and presumably formation of repressive higher order chromatin structure.

Nonetheless, it is clear, if our model is correct, that ZLD is fulfilling the same general role that pioneer factors carry out—getting to the genome first and facilitating the subsequent binding of other factors.

While there are no clear ZLD homologs outside of insects, it has recently been shown that in zebrafish the transcription factor Pou5f1 (Oct4), in combination with Nanog and SoxB1, drives zygotic genome activation and may share with ZLD a pioneer-like activity (*Lee et al., 2013*; *Leichsenring et al., 2013*). Together these data suggest that pioneer transcription factors may generally be required to prepare the embryonic genome for widespread transcriptional activation at the MZT. Interestingly, Pou5f1 is homologous to the canonical pluripotency factor Oct4, which along with Nanog and Sox2, are transcription factors expressed to generate induced pluripotent stem cells. Together these data make an explicit connection between the role of pioneer-like factors in zygotic genome activation and the establishment of a totipotent state.

## Separation of enhancer specification from output

Another attractive feature of the model presented above is that it would explain an important and unexplained question about transcriptional enhancers: given that essentially every enhancer sized stretch of the *Drosophila* genome contains a large number of binding sites for the factors active in the cellular blastoderm (or any other stage of development) (*Berman et al., 2002*), why is it that only a small fraction of the genome functions as an enhancer?

It has long been thought that the difference between enhancers and the remainder of the genome is that enhancers do not simply contain binding sites, but rather have these sites in a particular configuration that leads to activation. However, the arrangement of binding sites within *Drosophila* enhancers is highly flexible (*Ludwig and Kreitman, 1995*; *Ludwig et al., 1998*, *2005*; *Hare et al., 2008*), and we have struggled to find any evidence for strong 'grammatical' effects in enhancer organization.

The data presented here and elsewhere on ZLD binding and activity support the alternative explanation that the specification of enhancer location and output are distinct processes carried out by specific sets of factors: pioneer factors like ZLD—that determine where an enhancer will be, by influencing the maturation of genomic chromatin, and more classical patterning factors that determine what the transcriptional output of the enhancer will be.

## Materials and methods

### Antibodies

The antibodies for histone H3, and various histone modifications were purchased from commercial sources as listed in *Table 1*.

### Fly strains

The zld[294] mutant and the ovo[D1] mutant lines used to obtain *zld* maternal mutant embryos using the FLP-DFS technique (*Chou and Perrimon, 1996*) have been described previously (*Liang et al., 2008*) and were obtained from the laboratory of Christine Rushlow at New York University.

### In vivo formaldehyde cross-linking of embryos, embryo sorting, and chromatin preparation

*D. melanogaster* flies were maintained in large population cages in an incubator set at standard conditions (25°C). Embryos were collected for 30 min, and then allowed to develop for 55, 85, 120 or 160 additional minutes before being harvested and fixed with formaldehyde. The fixed embryos were hand sorted in small batches using an inverted microscope to remove embryos younger or older than the targeted age range based on morphology of the embryos as previous described (*Harrison et al., 2011*). After sorting, embryos were stored at −80°C. After all collections were completed, the sorted embryos of each stage were pooled, and a sample of each pool were stained with DAPI. The ages of the embryos and their distribution in the two younger embryo pools (c7–9, and c11–13) were determined based on nuclei density of the stained embryos. The ages of embryos between c14a and c14c, both which were distinct from c13 based nuclei density, were determined based on morphology. 7.5, 0.7, 0.4, and 0.3 g of embryos at four different stages respectively, were used to prepare chromatin for immunoprecipitation following the $CsCl_2$ gradient ultracentrifugation protocol as previously described (*Harrison et al., 2011*).

### ChIP and sequencing

The chromatin obtained was fragmented to sizes ranging from 100 to 300 bp using a Bioruptor (Diagenode, Inc., Seraing, Belgium) for a total of processing time of 140 min (15 s on, 45 s off), with power setting at 'H'. Prior to carrying out chromatin immunoprecipitation, we mixed the chromatin from each sample with a roughly equivalent amount of chromatin isolated from stage 5 (mitotic cycle 14) *D. pseudobscura* embryos, and used about 2 µg of total chromatin (1 µg each of the *D. melanogaster* and *D. pseudobscura* chromatin) for each chromatin immunoprecipitation. The chromatin immunoprecipitation reactions were carried out as described previously (*Harrison et al., 2011*) with 0.5 µg anti-H4K5ac (07-327; Millipore, Billerica, MA), 0.5 µg of anti-H3K4me3 (ab8580; Abcam, Cambridge, United Kingdom), 0.5 µg of anti-H3K27ac (ab4729; Abcam), 1 µg of anti-H3 (ab1791; Abcam), 0.75 µg anti-H3K4me1 (ab8895; Abcam), 0.75 µg anti-H4K8ac (ab15823; Abcam), 1.5 µl of anti-H3K9ac (39,138; Activemotif), 0.75 µg anti-H3K18ac (ab1191; Abcam), 3 µg anti-H3K27me3 (07-449; Millipore), or 0.75 µg anti-H3K36me3 (ab9050; Abcam). The sequencing libraries were prepared from the ChIP and Input DNA samples using the Illumina (San Diego, CA) TruSeq DNA Sample Preparation kit following the manufacturer's instructions, and DNA was subjected to ultra-high throughput sequencing on a Illumina HiSeq 2000 DNA sequencers.

### Mapping sequencing reads to the genome, and peak calling

Sequenced reads were mapped jointly to the April 2006 assembly of the *D. melanogaster* genome [Flybase Release 5] and the November 2004 assembly of the *D. pseudoobscura* genome [Flybase Release 1.0] using Bowtie (*Langmead, 2010*) with the command-line options '-q −5 5 -3 5 -l 70 -n 2 -a -m 1 –best -strata', thereby trimming 5 bases from each end of the 100 base single reads, and keeping only tags that mapped uniquely to the genomes with at most two mismatches. Each read was extended to 130 bp based on its orientation to generate the ChIP profiles. We called peaks for each experiment using MACS (*Zhang et al., 2008*) v1.4.2 with the options '-- nomodel--shiftsize = 130', and used Input as controls.

## Data normalization

The addition of *D. pseudoobscura* chromatin prior to the chromatin immunoprecipitation provided us with a means to normalize the ChIP signals for each histone mark and for H3 between different stages. To normalize, we first determined the scaling factor needed to normalize the number of reads for *D. pseudoobscura* to 10 million, and scaled the signals of *D. melanogaster* ChIP profile in each sample using this factor. We then multiplied the scaled *D. melanogaster* signals by the ratio of *D. pseudoobscura* reads to *D. melanogaster* reads in the Input sample, which represents the relative amounts of chromatin of the *D. melanogaster* and the *D. pseudoobscura* in the starting chromatin samples used for the chromatin immunoprecipitation reactions.

## Overall dynamics of ChIP signals under enrichment peaks

Starting with peaks called by MACS as described above, we identified subpeaks by peaksplitter [http://www.ebi.ac.uk/research/bertone/software], and generated a consolidated list of subpeaks for each histone mark for all stages by joining each group of subpeaks that are within 200 bp into a single peak. We calculated the ChIP signal for each subpeak at each stage by summing the ChIP signal around a 500 bp window center around of each peak position in the normalized ChIP profile generated as described above. To show the overall trend of each histone mark, the range of the ChIP signal among all the subpeaks at each stage is shown as box plot.

## Gene classification according to transcription dynamics

Using our previous single-embryo RNA-seq data from Lott et al. (*Lott et al., 2011*), genes were classified as zygotic or maternal. We further divided the zygotic genes into four different groups based on their onset of zygotic expression (first time point with FPKM>1). This includes 107 genes whose onset of expression was around mitotic cycles 10–11 ('Early' group), 99 genes at cycles 12–13 ('Mid'), 143 genes at early cycle 14 ('Late'), and 99 genes during late cycle 14 ('Later'). The maternal group of genes was then compared against post-MBT polII ChIP (*Chen et al., 2013*) and split into Maternal/Zygotic genes that show in vivo promoter binding of polII ('Mat/Zyg') and a group of genes that show no polII binding ('Mat–only'). In addition, we used RNA-seq data (*Lott et al., 2011*) to define another class of non-expressed genes, showing no transcription from mitotic cycles 10 through the end of cycle14 ('Silent').

## Defining embryo blastoderm enhancers

A set of putative enhancers was defined based on the in vivo binding locations for early transcription, as measured previously by us using ChIP–chip (*MacArthur et al., 2009*). Here, we summed the raw ChIP–chip signal for 16 factors, including the A/P (Bicoid, Caudal, Hunchback, Giant, Krüppel, Knirps, Huckebein, Tailless, and Dichaete) and D/V (Dorsal, Snail, Twist, Daughterless, Mad, Medea, and Schnurri) regulators. We then identified all regions with cumulative signal over 20. This yielded 784 genomic regions, with an average length of 488 bp. These putative enhancers were then classified based on their position with regard to nearby genes, retaining only a set of 588 intergenic and intronic putative enhancers.

## Analysis of motif enrichment

Two methods were used to investigate the DNA motifs enriched around the peaks identified by MACS. First, 7mers enriched in the 2 kb sequences around the peaks for each experiment were identified by comparing the frequency of each 7mer to the 7mer distribution in randomly selected 2 kb sequences throughout the genome. The selection of the random sequences was restricted to the major chromosome arms excluding the heterochromatic sequences, and the distribution of the number of random sequences were set to match the distribution of peaks among different chromosome arms. The enrichment of the 7mers was ranked based on Z scores. In parallel, the motif enrichment analysis was also carried out using MEME (*Bailey and Elkan, 1994*) with motif length set at 6–10 and maximum number of motifs to be found at 10. In this case the sequences located in the 250–650 bp surrounding the maximum of the histone mark peak were used, and random sequences selected using the same criteria as the kmer enrichment analysis were used as negative control. The search was limited to the 150 top ranked peaks for each histone mark. After the candidate enriched motif was identified from these two methods, the motifs were used to map the enrichment around all the peaks by patser (*Hertz and Stormo, 1999*) using a ln(p-value) cutoff of −7.5, and with Alphabet set at 'a:t 0.3 c:g 0.2'.

## ChIP-seq in *zld* mutant embryos

To obtain embryos depleted of maternal *zld* RNA, the FLP-DFS technique was used. Briefly, *zld*[294], *FRT19A/FM7* (*Liang et al., 2008*) virgin females were crossed with *ovo*[D1],*hsFLP112,FRT19A/Y* (*Liang et al., 2008*) males. The larvae developed from embryos laid by females from these crosses were heat-shocked twice, each for 2 hr at 37°C, when they were between 24–48 hr, and between 48–72 hr old. Collection of the mutant embryos from the resulting female progeny, as well as the aging and fixation of the embryos was carried out following standard protocol as described above except that the collection period is 3 hr followed by 1 hr aging. The embryos were sorted to remove deformed post cycle 14 embryos. As a control, wild-type embryos were collected, treated in parallel and sorted to remove embryos older than stage 5. The ChIP-seq was carried out with the chromatin from the mutant and wild-type embryos using anti-H3, anti-H3K18ac, anti-H3K4me1, and anti-ZLD antibodies as described above.

## Data availability

All raw data are available at the GEO database under the accession number GSE58935. A genome browser with tracks from the data generated and analyzed here is available at the UCSC genome browser.

## Acknowledgements

This was funded by an HHMI Investigator award to MBE. TK is a member of the Israeli Center of Excellence (I-CORE) for Gene Regulation in Complex Human Diseases (Israel Science Foundation grant No. 41/11), and the Israeli Center of Excellence (I-CORE) for Chromatin and RNA in Gene Regulation (grant No 1796/12).

## Additional information

### Funding

| Funder | Grant reference number | Author |
|---|---|---|
| Howard Hughes Medical Institute | | Xiao-Yong Li, Jacqueline E Villalta, Michael B Eisen |
| Israel Science Foundation | 41/11 | Tommy Kaplan |
| Israel Science Foundation | 1796/12 | Tommy Kaplan |

The funders had no role in study design, data collection and interpretation, or the decision to submit the work for publication.

### Author contributions

X-YL, Conception and design, Acquisition of data, Analysis and interpretation of data, Drafting or revising the article; MMH, TK, MBE, Conception and design, Analysis and interpretation of data, Drafting or revising the article; JEV, Acquisition of data

## Additional files

### Major datasets

The following dataset was generated:

| Author(s) | Year | Dataset title | Dataset ID and/or URL | Database, license, and accessibility information |
|---|---|---|---|---|
| Li XY, Harrison MM, Villalta JE, Kaplan T, Eisen MB | 2014 | Histone modifications during the Drosophila maternal-to-zygotic transcition | http://www.ncbi.nlm.nih.gov/geo/query/acc.cgi?acc=GSE58935 | Publicly available at NCBI Gene Expression Omnibus. |

The following previously published datasets were used:

| Author(s) | Year | Dataset title | Dataset ID and/or URL | Database, license, and accessibility information |
|-----------|------|---------------|----------------------|-------------------------------------------------|
| Harrison MM, Li XY, Kaplan T, Botchan MR, Eisen MB | 2011 | Zelda binding in the early Drosophila melanogaster embryo marks regions subsequently activated at the maternal-to-zygotic transition | http://www.ncbi.nlm.nih.gov/geo/query/acc.cgi?acc=GSE30757 | Publicly available at NCBI Gene Expression Omnibus. |
| Lott SE, Villalta JE, Schroth GP, Luo S, Tonkin LA, Eisen MB | 2011 | Noncanonical compensation of zygotic X transcription in early Drosophila melanogaster development revealed through single-embryo RNA-seq | http://www.ncbi.nlm.nih.gov/geo/query/acc.cgi?acc=GSE25180 | Publicly available at NCBI Gene Expression Omnibus. |
| Chen K, Johnston J, Shao W, Meier S, Staber C, Zeitlinger J | 2013 | A global change in RNA polymerase II pausing during the Drosophila midblastula transition | http://www.ncbi.nlm.nih.gov/geo/query/acc.cgi?acc=GSE41703 | Publicly available at NCBI Gene Expression Omnibus. |

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
