## [Decision Letter]

Thank you for sending your work entitled “Establishment of regions of genomic activity during the *Drosophila* maternal to zygotic transition” for consideration at *eLife*. Your article has been favorably evaluated by Chris Ponting (Senior editor), a Reviewing editor, and 2 reviewers.

The Reviewing editor and the reviewers discussed their comments before we reached this decision, and the Reviewing editor has assembled the following comments to help you prepare a revised submission.

The manuscript presents interesting data that provide a unique view into how histone modifications accumulate at enhancers that are activated during the maternal-to-zygotic transition. The reviewers agree in principle that this is worthy of publication in *eLife*. However, two major concerns on the manuscript need to be addressed before further consideration.

1) The first is a concern raised by reviewer 2 on the normalization of datasets across different samples (here embryonic stages and wildtype versus Zelda-depleted embryos), when the samples are expected to strongly differ in the strengths and relevant features.

2) The second is a shared general concern of both reviewers that key aspects of existing literature are not cited and integrated into the text. Several conclusions drawn are already well established in the literature. These omissions detract from the balance and quality of the work and must be addressed. Specific examples are noted in the reviewers' opinions, which are included below to aid the authors in revising the manuscript.

*Reviewer #1*:

Li et al. perform a careful genome-wide analysis of various histone modifications during the maternal-to-zygotic transition in *Drosophila* embryos. Their major finding is that histone acetylation at many residues is present before the mid-blastula transition and that the presence correlates with sites of Zelda motifs and Zelda binding. In contrast, histone methylation, including H3K4me1, which is thought to be a general mark of enhancers, is not present until after widespread transcription begins during the mid-blastula transition.

Overall, the data and the analysis seemed to be of high quality and provide a unique view into how histone modifications accumulate at enhancers that are activated during the maternal-to-zygotic transition. Unfortunately, the interpretation of the data frequently does not take into account the current knowledge of the literature and thus many observations that the authors describe are either already published or are very speculative and not well supported. While the novelty of the manuscript lies in the enhancer analysis before transcription begins, the authors spend quite a deal on well-studied areas such as the role of H3K27me3 or the differences in nucleosomes and histone modifications at different gene classes (see below). These oversights unfortunately detract from the manuscript and leave me wondering what the novelty and significance of the presented findings is. Looking at the data, I do think that the manuscript has value but I would have expected a much more balanced paper (with the inclusion of appropriate citations and better emphasis on what's novel) for publication in *eLife*.

Findings that were already known and not acknowledged:

1) The authors write “Despite the generally assumed correlation between H3K4me3 at promoters and H3K36me36 at gene bodies, our data demonstrate that at early expressed genes transcription proceeds in the absence of these marks (Figure 5).”

It has become clear in the last few years that this general assumption is incorrect. Rach et al. (PLOS Genetics 2011) showed that epigenetic features such as histone modifications differ between promoter classes and this difference has also been observed between the five chromatin states observed by Bas van Steensel'd group in Kc cells. They show that H3K36me3 is specific for yellow chromatin (housekeeping genes) versus red (developmental genes). Finally, Chen et al. (eLife 2013) showed that genes expressed before the midblatula transition in *Drosophila* do not show H3K4me3. The latter paper is cited in the manuscript with regard to the Pol II data so I am surprised that the authors do not cite the paper in this context.

2) The authors write “Surprisingly, we observed NFRs at the promoters of maternally deposited genes at all time points (Figure 5), even though there is no evidence that genes in this set are transcribed either from expression data or ChIP with RNA polymerase.”

Actually, the traditional view is usually that promoters are naturally nucleosome-free. However, it is known from yeast and *Drosophila* and that regulated promoters tend to have a strong disposition for promoter nucleosomes. For example, it has been shown by Gilchrist et al. (Cell 2010) and Gaertner et al. (Cell reports 2012) that promoters with paused Pol II (thus developmental genes) tend to have a strong promoter nucleosome when Pol II is not present. Thus, the fact that maternal genes and developmental genes have a different natural nucleosome configuration at the promoter was already known.

3) The authors find that H3K27me3 was completely absent from maternal genes and they speculate in quite some detail about the “balanced state” of Polycomb-repressed genes as if nothing had been said about this before. The fact that H3K27me3 is specifically found at developmental genes both in *Drosophila* and mammals has been known for a long time (e.g. Schwartz 2006, Boyer 2006, Lee 2006) and the relationship between H3K27me3 and Pol II (the balanced state) is fairly complex and not accurately described in the manuscript.

Recommended reading (to name a few): Dellino (Mol Cell 2004); Stock et al. (Nat Cell Biol 2007); Enderle et al. (Genome Res 2011); Gaertner et al. (Cell reports 2012); Bonn et al. (Nat Genet 2012); Lehmann et al. (JBC 2012)

4) The authors propose that inconsistent results from reporter assays on Zelda enhancers stem from the fact that reporter lines usually have insulator sites and thus are not under the influence of Polycomb repression. This is certainly a good point but has previously been reported by Arnold at el. (Science 2013). In fact, Arnold et al. found this in S2 cells, suggesting that this is a general phenomenon and not specifically linked to Zelda.

Other major points:

The authors describe “within the broad H3k27me3 domains, the signal was almost completely absent in regions corresponding to ZLD-binding peaks” and speculate a fair bit about the significance. However, the simplest explanation for this is that ZLD-bound regions are nucleosome-depleted and thus do not have H3K27me3 signal. Looking at the average plots of other histone modifications on Figure 6, this dip is certainly observed for all other histone modifications, too (although the scale is different so it's hard to compare). The overlap between ZLD-bound regions and H3K27me3 regions as shown in Figure 8 is actually fairly low (<50%) and could be explained by the fact that both Zelda and H3K27me3 are preferentially found at developmental genes. I would predict that the same relationship would be found between any other early transcription factor and H3K27me3 regions. I do find it very conceivable, however, that the H327ac that accumulates at ZLD enhancers would prevent H3K27me3 (as described by Tie et al. Development 2009) but the data shown in the manuscript have not convinced me yet that this is visible in the genome-wide H3K27me3 data (I am also surprised that the authors do not cite this paper).

The authors propose that some enhancers that are bound by Zelda and have H3K27me3 are poised for future activation later in development. Unfortunately, this is only based on one gene and its published expression pattern. At the very least, one would like to see that the ZLD-bound region is indeed a late enhancer (e.g. based on Kwon et al. Nature 2014) and that this pattern can be observed for multiple enhancers. To my knowledge, early ZLD-bound regions are generally not active late so I would need better evidence to be convinced otherwise.

*Reviewer #2*:

How zygotic genomes are transcriptionally activated during early embryogenesis is an important open question. The authors determine the genomic distribution of histone H3 and different histone marks at 4 stages of embryogenesis using ChIP from tightly staged embryos. They integrate these datasets with previously published data on transcription and the binding of Pol II and the transcription factor Zelda. This allows the authors to investigate the dynamics of these chromatin features in general, at the transcription starting sites of different sets of genes (e.g. genes expressed during early embryogenesis, maternal genes, etc.), and around the locations of putative blastoderm embryo enhancers. Overall, this reveals that the genome appears relatively uniformly at mitotic cycle 8 and acquires the different histone modifications and nucleosome free regions only later between cycles 12 and 14. Correlations between histone modification patterns and Zelda binding strengthens the previous notion that Zelda is involved in zygotic genome activation and in defining embryonic enhancers. The authors repeat some of their experiments and analyses in embryos in which Zelda had been genetically depleted and found that the chromatin structure at intergenic position that bind Zelda in wildtype embryos was strongly affected while little effect was observed at TSS.

I find this an interesting and important paper that presents a time-course of how prominent chromatin features associated with transcription and transcriptional regulation are formed during the activation of the zygotic genome in early embryogenesis. I have only 1 major technical comment and a few suggestions that might help the authors to clarify a few sections of the text and figures.

Major technical comment:

The normalization of datasets across different samples (here embryonic stages and wildtype versus Zelda-depleted embryos) is an important issue, especially when the samples are expected to strongly differ in the strengths or even the presence of the studied features. The authors present an elegant solution: prior to the chromatin immunoprecipication, they mix their stage-specific *D. melanogaster* chromatin samples with a constant spike-in sample, *D. pseudoobscura* chromatin from stage 5 (mitotic cycle 14) embryos. As the associated DNA molecules differ characteristically in their sequences, the origin of the sequencing reads can be determined after deep sequencing, allowing the normalization of the different *D. melanogaster* samples to the constant spike-in *D. pseudoobscura* sample.

Judging from the description of the data normalization, this spike-in control might however have been used sub-optimally; the authors state that they normalize their data based on the number of reads, which can only ensure that the same number of reads (i.e. the same 'sequencing power') goes into the analysis of each sample. While this makes sense for RNA-seq data, it does not account for possible experimental variability in a ChIP experiment and the results are in my opinion difficult to interpret. In other words, the number of reads is unrelated and uninformative regarding the quality of a ChIP enrichment, i.e. if the IP step enriched for the respective antibody epitope comparably strongly or not. Similarly, it cannot control for potential differences in library complexity, i.e. the number of identical reads that can arise during PCR amplification.

To demonstrate that the different experiments worked comparably, the authors should determine peaks in the D. pseudoobscura spike-in sample and show that they are of similar heights across the different experiments (e.g. next to the respective *D. melanogaster* data). This is particularly relevant when no signal is seen in the *D. melanogaster* samples, e.g. for early stages in Figures 5 and 6, and the current normalization cannot exclude that the ChIP failed.

---

## [Author Response]

We have addressed the concern of reviewer 2 about normalization with several additional analyses, and we have extensively rewritten the manuscript to more fully and accurately cite the relevant literature and contextualize our results. We have also removed several sections that were too speculative and superfluous to our main point.

Reviewer #1:

[…]

*1) The authors write “Despite the generally assumed correlation between H3K4me3 at promoters and H3K36me36 at gene bodies, our data demonstrate that at early expressed genes transcription proceeds in the absence of these marks (*Figure 5*)*.*”*

*It has become clear in the last few years that this general assumption is incorrect. Rach et al. (PLOS Genetics 2011) showed that epigenetic features such as histone modifications differ between promoter classes and this difference has also been observed between the five chromatin states observed by Bas van Steensel's group in Kc cells. They show that H3K36me3 is specific for yellow chromatin (housekeeping genes) versus red (developmental genes). Finally, Chen et al. (eLife 2013) showed that genes expressed before the midblatula transition in* Drosophila *do not show H3K4me3. The latter paper is cited in the manuscript with regard to the Pol II data so I am surprised that the authors do not cite the paper in this context.*

We have modified the text referred to here (which is from the Results section) to simply note the absence of these methyl marks at transcribed genes prior to the cellular blastoderm. We have expanded the discussion to properly situate it with respect to earlier observations.

*2) The authors write “Surprisingly, we observed NFRs at the promoters of maternally deposited genes at all time points (*Figure 5*), even though there is no evidence that genes in this set are transcribed either from expression data or ChIP with RNA polymerase*.*”*

*Actually, the traditional view is usually that promoters are naturally nucleosome-free. However, it is known from yeast and* Drosophila *and that regulated promoters tend to have a strong disposition for promoter nucleosomes. For example, it has been shown by Gilchrist et al. (Cell 2010) and Gaertner et al. (Cell reports 2012) that promoters with paused Pol II (thus developmental genes) tend to have a strong promoter nucleosome when Pol II is not present. Thus, the fact that maternal genes and developmental genes have a different natural nucleosome configuration at the promoter was already known.*

Here too we have modified the text in the Results to simply point out that the non-transcribed maternal genes have a clear nucleosome-free region, and to highlight the earlier work from Gaertner (which we think is more relevant to our point than Gilchrist). We point out the new contribution from our data, which is the developmental stability of the maternal-gene associated NFR, and interpret it as consistent with the observation from Gaertner.

*3) The authors find that H3K27me3 was completely absent from maternal genes and they speculate in quite some detail about the “balanced state” of Polycomb-repressed genes as if nothing had been said about this before. The fact that H3K27me3 is specifically found at developmental genes both in* Drosophila *and mammals has been known for a long time (e.g. Schwartz 2006, Boyer 2006, Lee 2006) and the relationship between H3K27me3 and Pol II (the balanced state) is fairly complex and not accurately described in the manuscript*.

*Recommended reading (to name a few)*:

*Dellino (Mol Cell 2004); Stock et al. (Nat Cell Biol 2007); Enderle et al. (Genome Res 2011); Gaertner et al. (Cell reports 2012); Bonn et al. (Nat Genet 2012); Lehmann et al*. *(JBC 2012)*

We have removed this discussion of the balanced state from the Results. It was superfluous to our main point and, as the reviewer points out, our data do not significantly enhance our understanding of this state. We thank the reviewer for the guidance and added relevant references.

*4) The authors propose that inconsistent results from reporter assays on Zelda enhancers stem from the fact that reporter lines usually have insulator sites and thus are not under the influence of Polycomb repression. This is certainly a good point but has previously been reported by Arnold at el. (Science 2013). In fact, Arnold et al. found this in S2 cells, suggesting that this is a general phenomenon and not specifically linked to Zelda*.

We did not mean to suggest that it was a new idea that enhancers can be silenced in their endogenous location but active in a transgene assay. Rather, we were suggesting a specific mechanism to explain the observed effect. However, this is really just speculation that could be relatively easily tested, so we have removed this section of the manuscript and will investigate this question experimentally and publish it elsewhere.

*Other major points*:

*The authors describe “within the broad H3k27me3 domains, the signal was almost completely absent in regions corresponding to ZLD-binding peaks” and speculate a fair bit about the significance. However, the simplest explanation for this is that ZLD-bound regions are nucleosome-depleted and thus do not have H3K27me3 signal. Looking at the average plots of other histone modifications on*
Figure 6*, this dip is certainly observed for all other histone modifications, too (although the scale is different so it's hard to compare)*.

We agree that the simplest explanation for the drop in H3K27me3 levels in the immediate vicinity of ZLD is nucleosome depletion, although the regions of H3K27me3 depletion tend to be larger than that of nucleosome depletion. However, our primary point was that ZLD binding tends to interrupt regions H3K27me3 – it is this spatial relationship, not the specific mechanism of depletion, to which we were referring. We have rewritten the sections referring to H3K27me3 to make this clear.

*The overlap between ZLD-bound regions and H3K27me3 regions as shown in*
Figure 8
*is actually fairly low (<50%) and could be explained by the fact that both Zelda and H3K27me3 are preferentially found at developmental genes. I would predict that the same relationship would be found between any other early transcription factor and H3K27me3 regions*.

We have dropped this section from the paper. We agree with the reviewer, it was not compelling.

*I do find it very conceivable, however, that the H327ac that accumulates at ZLD enhancers would prevent H3K27me3 (as described by Tie et al. Development 2009) but the data shown in the manuscript have not convinced me yet that this is visible in the genome-wide H3K27me3 data (I am also surprised that the authors do not cite this paper)*.

We have added a discussion of Tie’s results to the Discussion section, and agree that this is a tantalizing possibility, although our current data do not really speak to it.

*The authors propose that some enhancers that are bound by Zelda and have H3K27me3 are poised for future activation later in development. Unfortunately, this is only based on one gene and its published expression pattern. At the very least, one would like to see that the ZLD-bound region is indeed a late enhancer (e.g. based on Kwon et al. Nature 2014) and that this pattern can be observed for multiple enhancers. To my knowledge, early ZLD-bound regions are generally not active late so I would need better evidence to be convinced otherwise*.

There are several additional examples of this, but we feel that additional data are needed to understand their functional relevance, so we have removed any discussion of poised enhancers from the paper.

Reviewer #2:

*[…] Judging from the description of the data normalization, this spike-in control might however have been used sub-optimally; the authors state that they normalize their data based on the number of reads, which can only ensure that the same number of reads (i.e. the same 'sequencing power') goes into the analysis of each sample. While this makes sense for RNA-seq data, it does not account for possible experimental variability in a ChIP experiment and the results are in my opinion difficult to interpret. In other words, the number of reads is unrelated and uninformative regarding the quality of a ChIP enrichment, i.e. if the IP step enriched for the respective antibody epitope comparably strongly or not. Similarly, it cannot control for potential differences in library complexity, i.e. the number of identical reads that can arise during PCR amplification*.

*To demonstrate that the different experiments worked comparably, the authors should determine peaks in the* D. pseudoobscura *spike-in sample and show that they are of similar heights across the different experiments (e.g. next to the respective* D. melanogaster *data). This is particularly relevant when no signal is seen in the* D. melanogaster *samples, e.g. for early stages in*
Figures 5 and 6*, and the current normalization cannot exclude that the ChIP failed.*

We agree with everything the reviewer says here. Indeed we had done all of the analyses they suggest in the early stages of this paper, and chose to use the simple read-based normalization because the results were the same as when we used peak heights. We note that while it is certainly true that there is a lot of variation between antibodies in the efficacy of the immunoprecipitation, here we are only normalizing between parallel IPs using the same antibody (and carried out at the same time). While it is obviously possible for there to be differences in IP efficiency between experiments with the same antibody, our results here, and more generally, suggest that this is relatively uncommon, meaning that read- based normalization will work in most cases. Nonetheless, we have added an additional figure addressing consistency in the *D. pseudoobscura* IPs and discuss the use of peak-based normalization in the text.